# CAR-T Cell Therapy in the Treatment of Pediatric Non-Hodgkin Lymphoma

**DOI:** 10.3390/jpm13111595

**Published:** 2023-11-10

**Authors:** Magdalena Ostojska, Emilia Nowak, Julia Twardowska, Monika Lejman, Joanna Zawitkowska

**Affiliations:** 1Student’s Scientific Association of the Department of Pediatric Hematology, Oncology and Transplantation, Medical University of Lublin, 20-093 Lublin, Poland; magdalenaostojska.mo@gmail.com (M.O.); emilia.m.nowak@wp.pl (E.N.); julka.twardowska@gmail.com (J.T.); 2Independent Laboratory of Genetic Diagnostics, Faculty of Medicine, Medical University of Lublin, 20-093 Lublin, Poland; monikalejman@umlub.pl; 3Department of Pediatric Hematology, Oncology and Transplantation, Medical University of Lublin, 20-093 Lublin, Poland

**Keywords:** non-hodgkin lymphoma, CAR-T cells, pediatric patients, marker, CD19, CD20, CD22, immunotherapy, adverse effects, limitations

## Abstract

Non-Hodgkin lymphomas (NHL) are a group of cancers that originate in the lymphatic system, especially from progenitor or mature B-cells, T-cells, or natural killer (NK) cells. NHL is the most common hematological malignancy worldwide and also the fourth most frequent type of cancer among pediatric patients. This cancer can occur in children of any age, but it is quite rare under the age of 5 years. In recent decades, available medicines and therapies have significantly improved the prognosis of patients with this cancer. However, some cases of NHL are treatment resistant. For this reason, immunotherapy, as a more targeted and personalized treatment strategy, is becoming increasingly important in the treatment of NHL in pediatric patients. The objective of the following review is to gather the latest available research results, conducted among pediatric and/or adult patients with NHL, regarding one immunotherapy method, i.e., chimeric antigen receptor (CAR) T cell therapy. We focus on assessing the effectiveness of CAR-T cell therapy, which mainly targets B cell markers, CD19, CD20, and CD22, their connections with one another, sequential treatment, or connections with co-stimulatory molecules. In addition, we also evaluate the safety, aftermath (especially neurotoxicities) and limitations of CAR-T cell therapy.

## 1. Introduction

Non-Hodgkin lymphoma (NHL) is one of the most common childhood malignancies. Its wide occurrence is undoubtedly connected to the heterogeneous nature of NHL. The classification of NHL is most frequently based on immunophenotype (i.e., B-lineage, T-lineage) and molecular biology. The majority of pediatric NHL cases represent three types: lymphoblastic lymphoma, anaplastic large cell lymphoma, as well as aggressive mature B-cell NHL (B-NHL), which primarily includes Burkitt lymphoma (BL), diffuse large B-cell lymphoma (DLBCL) and, less common than the other two subtypes, primary mediastinal B-cell lymphoma. Other NHL types, such as pediatric gray zone lymphoma, marginal zone lymphoma, primary central nervous system (CNS), peripheral T-cell lymphoma lymphoma or cutaneous T-cell lymphoma are much rarer [1]. Table 1 collects main types of pediatric NHL.

In 2020, the NHL incidence rate per 100,000 children younger than 20 years in the United States was 25.4 [2]. This accounted for approximately 45% of all pediatric lymphomas and 14% of all malignant tumors in children and teenagers. Interestingly, NHL prevalence varies depending on the age of the patient. In the 0–14 years age group, the dominant NHL type is BL with patients’ average age of 6 when diagnosed [3]. Annually, the incidence is estimated at 3 to 6 cases per 100,000 [4]. Among adolescents between 15 to 19 years old, the primary NHL is DLBCL with an incidence rate of 7 per 100,000 in the USA [3,5]. Boys tend to have higher risks of NHL than girls for most lymphoid neoplasms [3]. Figure 1 and Figure 2 present the percentage of respective NHL types in childhood and adolescence.

Such an extended group requires a broad range of treatment. Thus far, a combination of systemic chemotherapy is invariably recommended for the majority of patients, both newly diagnosed and with recurrent NHL [6]. Surgical treatment, if available, is mostly dedicated for patients with early-stage NHL, or for diagnosing the tumor. In contrast to adult patients, the use of radiotherapy is limited, as it is considered a risk factor of consecutive tumors, progressive disease, or chronic health conditions [7]. Although overall survival (OS) rates rose to 90% in the course of the past three decades, outcomes for patients with NHL remain unsatisfactory, mostly due to not only therapy-related morbidity, but poor health-related quality of life as well. Furthermore, unproductive dosage increase contributed to enhancement of adverse effects and resistance to hitherto therapy [8]. Lastly, even 30% of pediatric patients are likely to relapse. All these factors led to acceleration in the search for more effective medications [9].

The pursuit turned out to be successful as it resulted in immunotherapy findings, such as monoclonal antibodies (mAbs), various types of inhibitors, e.g., immune checkpoint inhibitors (ICIs), antibody–drug conjugates (ADCs), as well as chimeric antigen receptor (CAR) T-cell. In CAR-T cell therapy, T lymphocytes with a synthetic CAR are utilized for identifying and subsequently erasing specific tumor cells, regardless of major histocompatibility complex (MHC) molecules [10]. The targets for CAR-T cells are principally B-cell markers CD19, CD20, and CD22, present in numerous B-cell malignancies. Unfortunately, in consequence, both cancerogenic and normal B-cells are eliminated. Therefore, in order to prevent this process, alterations were introduced, which resulted in the creation of newer and newer generations of CAR-T cells.

## 2. The Possibility of Using CAR-T Cells in NHL Therapy

### 2.1. Mechanism of Action of CAR-T Cells

A CAR construct can bind to target markers through a single-chain variable fragment (scFv) recognition domain. By means of an extracellular link and a transmembrane domain, it is connected to an intracellular molecule called cluster of differentiation (CD)-3 3ζ chain in order to stimulate cell activation [11,12]. Such a formation forges a first-generation CAR. The T cells used in CAR-T therapy are genetically engineered with a T cell receptor (TCR) and its CD3ζ domain incorporates three immuno-tyrosine activation motifs (ITAMs) [13]. Lymphocyte-specific protein tyrosine kinase (Lck)-mediated phosphorylation of ITAMs inaugurates a signal within the cytoplasm.

Over the years, advances in the existing CAR structure have been made, resulting in the creation of second and third generations of CAR molecules. What distinguishes them from first generation CARs is the presence of different intracellular domains involved in the signaling process, such as 4-1BB, CD28, or inducible T cell co-stimulator (ICOS) [14,15]. Introduction of signaling domains from cytokine receptors and inducible expression of inflammatory cytokines, e.g., interleukin 12 (IL-12) or IL-18, led to the development of fourth and fifth generation CAR-T [16,17].

### 2.2. CAR-T Cells Anti-CD19

CD19 is a type 1 transmembrane glycoprotein, composed of a cytoplasmic C-terminus, an extracellular N-terminus, and a transmembrane domain in between. It is associated with the immunoglobulin family. CD19 is expressed on the surface of B lymphocytes regardless of their stage of maturation and differentiation. CD19 is also present on cells that have transformed into cancer cells. It is found on cancer cells in B-NHL in more than 95% of cases. Therefore, CD19 has become a suitable target for the treatment of these lymphomas, using CAR technology [18,19]. To date, the United States Food and Drug Administration (FDA) and/or the European Medicines Agency (EMA) have approved four CD19-targeted CAR-T cell drugs for only adult patients with B-NHL. Their brief characteristics are presented in Table 2. Of these, only tisagenlecleucel has been approved for the treatment of pediatric patients with relapsed or refractory ALL. Currently, no CD19-targeted CAR-T cell drug has been approved for pediatric NHLs. However, there are only a few studies (completed or currently recruiting patients) on this topic [20].

#### 2.2.1. Axicabtagene Ciloleucel (Other Terms: Axi-Cel, Yescarta, KTE-CD19)

The pivotal multicenter phase 1–2 trial leading to the approval of axi-cel for the treatment of relapsed or refractory (R/R) NHL was ZUMA-1, which included 101 patients aged 23–76 with treatment-resistant lymphoma. The overall response rate (ORR) was 74%, with as many as 54% of patients achieving complete response (CR). In 11 of 33 patients, a partial response (PR) in the first month after infusion converted to CR (usually up to 6 months). The median duration of response (DOR) and the median progression-free survival (PFS) were 11.1 and 5.9 months, respectively. The reduced and less durable response to axi-cel was shown to be associated with higher initial tumor mass, high level of inflammatory markers and lower CAR-T cell expansion in vivo after infusion. Axi-cel was characterized by a manageable long-term safety profile. A median OS exceeded 2 years [21,22,23]. In the primary analysis of the ZUMA-5 study, 104 adult people with R/R indolent NHL were administered a single infusion of axi-cel (2 × 10^6^ CAR-T cells per kg). The ORR was 92% of patients, including 74% with a CR. In the updated study population analysis of 109 people, these values were 94% and 79%, respectively. And more favorable treatment effects were obtained in patients with follicular lymphoma (FL) than marginal zone lymphoma. However, CAR-T cell expansion was higher in the second case. With a median follow-up of 23.3 months, responses to axi-cel were durable. Lymphoma progression occurred in 26% of patients with an initial response to treatment [24].

In the study by Neelapu et al. (ZUMA-12), axi-cel can be one of the effective first-line drugs in patients with high-risk large B-cell lymphoma (LBCL). Analysis of 37 patients aged 23–86 showed that the objective response rate (ORR) and complete response rate (CRR) were 89% and 78%, respectively. The median time to complete a response was 30 days. Additionally, at a median follow-up of 15.9 months, the objective response rate was maintained in 73% of participants. The median time to peak CAR-T cells expression was 8 days [25]. Seitter et al. reported the cases of two men, 33 and 45 years old, with aggressive BL treated with axi-cel. A single infusion of CAR-T cells was enough to achieve long-term remission. Patients had manageable cytokine release syndrome (CRS) and neurologic events [26].

However, currently no clinical trial results assessing the efficiency and safety of CAR-T therapy based on axi-cel in pediatric NHL are available. And because of the beneficial therapeutic potential of axi-cel in adults, clinical trials among children would be desirable.

#### 2.2.2. Tisagenlecleucel (Other Terms: Tisa-Cel, Kymriah, CTL019)

JULIET was the most important multicenter phase 2 trial that aimed to assess the efficacy and safety of tisa-cel therapy in the treatment of adults with R/R DLBCL, high-grade B-cell lymphoma (HGBCL), or transformed follicular lymphoma (tFL). The study recruited 115 patients aged 22–76 years. The ORR was 52%. And 40% of the trial participants achieved a CR on tisa-cel therapy. Moreover, in approximately 54% patients the initial PR to tisa-cel therapy was finally converted to CR. The effectiveness of tisa-cel treatment did not depend on the patient’s age, gender, initial general condition, tumor size, or previously used treatment methods [21,22]. The study also examined long-term clinical outcomes of tisagenlecleucel. The median follow-up of 115 adult patients was 40.3 months. After this time, a CR or PR to treatment was observed in 39% and 14% of patients, respectively. Furthermore, research proved that CR to this treatment at 3 or 6 months is a reliable early indicator of long-term survival. A total of 70 of 115 patients died, but the deaths were not related to tisagenlecleucel. Significant differences were not found between short- and long-term analyses of the safety profile of tisa-cel therapy. JULIET demonstrated the durability of response to tisa-cel and its favorable safety profile [27]. Cellular kinetics analysis showed that in patients with R/R DLBCL, tisa-cel reached maximum transgene levels at approximately 9 days after infusion, which persisted for up to 24 months (maximum of 693 days by quantitative polymerase chain reaction and 554 days by flow cytometry). In the blood of treatment nonresponders, tisa-cel persisted for a maximum of 374 and 400 days, depending on the diagnostic method [28].

In the study by Schuster et al., a complete remission (CR) of 6 months after tisa-cel infusion was achieved in 6 of 14 adults with DLBCL and in 10 of 14 adults with FL. There were high rates of durable remission of 7.7 to 37.9 months (median = 29.3 months). A median peak tisa-cel expansion in the blood occurred after 8 days in patients with a response and after 10 days in nonresponders [29]. Schuster et al. conducted a prospective clinical trial to confirm the effectiveness of tisa-cel therapy in 93 adult patients with R/R DLBCL. In those patients, autologous hematopoietic stem-cell transplantation (auto-HSCT) was ineffective or they did not meet the requirements for this transplantation. CR or PR to treatment occurred in 52% of the participants (40% and 12%, respectively). After 3 months, the treatment response rate was 32% for CR and 5% for PR. What is more, at 12 months, the researchers estimated that the rate of relapse-free survival could be approximately 65% of all the patients and 79% among patients who achieved a CR. No statistically significant differences between treatment responses in patients with different tumor CD19 expression were identified. The said response rates were stable for 6 months [30]. Tisa-cel infusions were also administered in eight patients aged 17–79 with secondary CNS lymphoma after 3–5 prior lines of treatment. This therapy had a favorable safety profile. Patients developed neurotoxicity of a maximum of grade 1 [31]. The ELARA global study evaluated the effectiveness of tisa-cel in nine adult patients aged 47–71 with R/R FL and after a minimum of two prior lines of treatment, or after auto-HSCT and relapse. At the beginning seven out of nine patients had stage III–IV disease, according to the Ann Arbor classification. Additionally, all patients received lymphodepleting therapy, and seven of nine patients received bridging therapy. CRR was 100%. One of the reasons was that tisa-cel was an early treatment line in those patients. Only one patient with high-risk baseline lymphoma had lymphoma relapse 289 days after tisa-cel infusion. In general, tisa-cel therapy had a manageable safety profile [32].

So far, the only study examining the efficacy and safety of tisa-cel in the treatment of lymphomas conducted in the pediatric population with CD19+ R/R B-cell non-Hodgkin’s lymphoma was the BIANCA study completed in 2023. Currently, only preliminary results of this study are available. Eight patients aged 9 to 16 years were recruited. They were diagnosed with LBCL (*n* = 4), BL (*n* = 3), and gray zone lymphoma (*n* = 1). A total of five of eight patients were treated with two or more lines of therapy before entering the study. The efficacy of tisa-cel was evaluated through ORR, which included both CRs and PRs. Pre-study treatment included bridging chemotherapy and lymphodepleting chemotherapy (fludarabine and cyclophosphamide). Then, each child received a single intravenous infusion of tisagenlecleucel. The dose depended on the patient’s body weight and was 0.9–1.7 × 10^6^ CAR + viable T cells/kg. The maximum transgene level observed in peripheral blood after single dose administration of tisa-cel did not exceed those observed in acute lymphoblastic leukemia (ALL) in children and young adults. The CRS and neurologic events occurred in five and three patients, respectively. Additionally, due to the progress of lymphoma, one death was reported. Currently, the study has expanded to 35 participants, but results are not available yet [33,34].

#### 2.2.3. Brexucabtagene Autoleucel (Other Terms: Tecartus, KTE-X19)

Successful results of the multicenter phase 2 ZUMA-2 trial in 2020 contributed to the registration of KTE-X19 in the treatment of R/R mantle cell lymphoma. Before the single infusion of KTE-X19 (2 × 10^6^ CAR-T cells per kg of body weight), 74 patients aged 38–79 received leukapheresis, bridging therapy (only optional) and conditioning chemotherapy. An objective response was observed in 85% patients and 59% of them had a CR. The median time to obtain an initial response and CR was 1 and 3 months, respectively. A total of 24 of 42 patients with an initial PR or stable disease progressed to a CR. After 12 months of follow-up, OS and PFS were 83% and 61%, in sequence [35].

There is also an ongoing study ZUMA-4 on the efficacy and safety of KTE-X19 in pediatric and adolescent participants aged 1–21 with R/R B-precursor ALL or B-NHL. In this clinical trial, patients initially receive conditioning chemotherapy (fludarabine and cyclophosphamide) followed by a single infusion of KTE-X19 at a dose of 2 × 10^6^ or 1 × 10^6^ anti-CD19 CAR-T cells per kg. The primary endpoints are: the overall CRR in ALL patients, the objective response rate in NHL patients, and the percentage of children experiencing adverse events (dose-limiting toxicities, DLT). The duration of patient observation is 2 years. Then, the patients will be transferred to another long-term study whose aim will be to finish/complete the 15-year follow-up assessments [36].

#### 2.2.4. Lisocabtagene Maraleucel (Other Terms: Liso-Cel, Breyanzi, JCAR017)

The pivotal study for liso-cel was called TRANSCEND. In its phase 1, researchers assessed liso-cel therapy in 23 patients aged 50–80 with R/R chronic lymphocytic leukemia (CLL)/small lymphocytic lymphoma (SLL). Patients were divided into two groups, depending on dose (50 or 100 × 10^6^ CAR-T cells). Overall response and CR occurred in 82% and 45% of the patients. The safety and efficacy result of liso-cel allowed the study to continue [37]. Phase 2 TRANSCEND evaluated the effectiveness of liso-cel in 269 patients aged 18–86 with DLBCL and relapse or progression after ≥2 prior lines of therapy (including prior allogeneic stem cell transplantation or secondary CNS involvement). Patients were divided into three groups depending on the dose of liso-cel (50 or 100 or 150 × 10^6^ CAR-T cells). With the median follow-up of 18.8 months, the ORR was 73%, which included 53% of patients achieving a CR. Those rates did not differ by dose level. The response to treatment was worse among patients with larger tumor diameter or elevated lactate dehydrogenase (LDH). The median time to achieve the first CR or PR was 1 month (range 0.7–8.9 months). A conversion of PR to CR occurred in 28 patients, with a median time of 3 months [38]. Basing on TRANSCEND, Patrick et al. used the European Organization for Research and Treatment of Cancer Quality of Life Questionnaire (EORTC QLQ-C30) to analyze the population consisting of 181 patients at a mean age of 60.2. The researchers demonstrated that liso-cel (administered as third-line or later treatment) improved health-related quality of life (HRQoL) and symptoms in patients with R/R LBCL. In addition, clinically meaningful improvements of global health status/Quality of Life (QoL) occurred in a higher proportion of treatment responders, compared with nonresponders (72% vs. 41%) [39]. The phase 3 TRANSFORM trial compared liso-cel therapy with standard of care (salvage chemotherapy followed by autologous stem cell transplantation) as second-line treatment in patients with R/R LBCL. Each group (depending on the type of therapy) consisted of 92 patients aged 42–67. The median follow-up time was 6.2 months. The ORR and CRR therein were higher in the liso-cel group (86% and 66%), compared to the control group treated with standard therapy (48% and 39%). However, the median time to achieve a CR was shorter in the standard group than in the liso-cel group (14.5 months vs. median not reached). In turn, PFS significantly improved after liso-cel therapy, compared with standard therapy [40].

Segal et al. evaluated the effectiveness of liso-cel as second-line therapy (after CD20-targeted therapy) in 61 adults aged 70–78 with Positron-Emission Tomography-positive (PET-positive) R/R LBCL not eligible for hematopoietic stem cell transplantation (Phase 2 PILOT). All patients underwent leukapheresis, lymphodepleting chemotherapy and next a liso-cel infusion. The median follow-up after infusion of liso-cel was 12.3 months. A total of 80% of the participants achieved an overall response, with 54% achieving a CR. In 13% of the patients (*n* = 8), the initial PR converted to a CR. The median DOR in the participants who achieved CR or PR was 12.09 months and it was 21.65 months for CR. The median time to first CR was approximately 1 month [41].

Similarly to axi-cel, no clinical trial results on the effectiveness and safety of liso-cel therapy in NHL children and/or adolescents have been published yet.

#### 2.2.5. CAR-T Cells in Burkitt Lymphoma after Liver Transplantation

Wang et al. describe the case of a patient aged 2 years 10 months who developed refractory BL (a rare form of post-transplant B-cell lymphoproliferative disorder, PTLD) after liver transplantation (due to decompensated cholestatic cirrhosis). The child received an autologous anti-CD19 CAR-T cells infusion at the total dose of 9.0 × 10^6^ per kg. The child developed a high fever (days 2–5 post-infusion), significant increases in serum inflammatory cytokines (particularly IL-6 on day 4 after infusion) and manageable CRS. After 55 days, the Positron-Emission Tomography and Computed Tomography (PET-CT) scan did not detect any previously existing cancer lesions in the abdominal cavity, pelvis or skeleton. A CR was achieved. Follow-up CT scans performed over the next 16 months did not depart from the norm, either. The therapy proved to be an effective and safe therapeutic option in PTLD, especially BL [42].

### 2.3. Other CAR-T Cells against NHL

Despite the impressive efficacy of CD19 CAR-T therapy, progressive disease occurs in a large proportion of patients who receive a CAR T-cell infusion, primarily as a result of a lack of CAR T-cell persistence and tumor cell resistance stemming from antigen loss or reduced antigen expression below the threshold required for CAR T-cell activity [43]. Sometimes mutations in the CD19 antigen and the downregulation or disappearance of this antigen from the surface of malignant lymphocytes lead to tumor escape [44]. The escape mechanisms associated with antigen loss in B-ALL during CD19 CAR T-cell therapy include alternative splicing of CD19, frameshift mutations, and missense mutations [43]. Because of these mechanisms some patients become resistant to CD19-targeted CAR-T treatment. Therefore, alternative markers, such as CD20 and CD22, with a higher expression in B-NHL can be used as targets for T cell therapies [44].

#### 2.3.1. Anti-CD20

CD20 is a non-glycosylated membrane phosphoprotein that is highly expressed both in normal B cells and on the surface of malignant B cells [44]. Moreover, the overexpression of CD20 could indicate highly progressive disease [20]. It is not found on hematopoietic stem cells, making them one of the most promising treatment targets for B-cell malignancies [45]. Compared with CD19, CD20 is much more slowly endocytosed following antibody binding and this stability could theoretically positively affect the quality of the immunological synapse, resulting in more robust CAR triggering and T-cell activation [46,47]. Unfortunately, clinical trial results assessing the efficiency of CD20-targeted CAR-T therapy in pediatric NHL are not available.

Recently, a phase I clinical trial studied the effectiveness and toxicity of CD20 CAR-T cells as salvage therapy in 15 adult R/R B-NHL patients previously treated with rituximab. The study revealed the ORR of 100% with 80% (12/15) CRs and 20% (3/15) partial remissions (PRs). All enrolled patients showed an improved clinical response, with a 3-month ORR up to 86.7%, with 46.7% patients (7/15) achieving CR and 40% patients (6/15) achieving PR for the best response. Moreover, half of the enrolled patients had achieved sustained CR lasting 10.3 months. Additionally, Cheng et al. showed encouraging results for CD20 CAR T cells in rituximab-refractory CD20+ B-NHL patients with CNS involvement, indicating that although CD20 antigen loss and downregulation have been described following CD20-based antibody therapy, CD20 CAR-T cells could still provide an alternative target that would allow sequential therapy [47].

#### 2.3.2. Bispecific and Dual Targeting CAR-T Cells

A bispecific receptor consists of two distinguished antigen recognition domains that bind to two separate intracellular domains and are expressed as tandem scFvs in one CAR, or as two different CARs on T cell surface [48]. If one of the target molecules is not available to CAR-T cells for reasons such as removal or mutation of the target antigen on malignant cells, a dual-function machine can largely prevent tumor escape. Thus, the bispecific CAR retains the cytolytic property of T cells [49].

##### CD19/CD20

CD19/CD20-bispecific CAR-T cells have been presented as a new synthetic molecule that, after recognition and binding to target tumor antigens on the surface of malignant cells, can establish a synergistic cascade of executive molecules [48]. Unfortunately, clinical trial results assessing the efficiency of CD20-targeted CAR-T therapy in pediatric NHL are not available.

The phase I/IIa clinical trial by Tong et al. was designed to evaluate the safety, efficacy and feasibility of administering tandem single-chain bispecific CD19/CD20 CAR-T cells (TanCAR7 T) to 28 adult patients with R/R B-cell lymphoma. In this study, the best ORR was 79%. A total of 71% of the patients achieved a CR and 7% had a PR. The PFS rates were 79% at 6 months and 64% at 12 months. Among the patients who achieved a CR at 3 months, the estimated PFS was 85% at 12 months. Among the patients with DLBCL, 75% (12/16) had a response, the median PFS was not reached, and 75% showed no disease progression at 12 months after infusion. The median OS was not reached, with OS rates of 82% at 6 months and 71% at 12 months. Among the patients who had a CR at 3 months, the estimated OS rate was 90% at 12 months. Of the four patients who were CD19^−^ at baseline by flow cytometry and immunohistochemistry, two achieved a CR as the best response, one had a PR and one showed PD. This study revealed that TanCAR7 T cells with dual antigen coverage of CD19 and CD20 form superior and stable immunological synapse structures, which may be related to the stronger and more rapid degranulation observed in TanCAR7 T cells than in single-targeted CAR-T cells. The cytolytic degranulation of perforin and granzyme is considered the main mechanism by which CAR-T cells redirect the killing of target cells [43].

The results of the Shah et al. phase I clinical trial of bispecific anti-CD20 and anti-CD19 (LV20.19) CAR-T in adult patients with B-NHL or CLL also confirmed the efficiency and safety of that therapy. A total of 82% (18/22) of the patients achieved an overall response at day 28, 64% (14/22) had a CR, and 18% (4/22) had a PR. Four patients had evidence of progressive disease by day 28. Among the patients who received the highest dose (2.5 × 10^6^ cells per kg; n = 16), the ORR at day 28 was 88%, with a CR rate of 75% (12/16). Among the patients who achieved a CR by day 28, the median DOR was not reached, with a median follow-up time of 10.1 months, while those who achieved a partial remission (PR) had a median DOR of 2.3 months. The median OS for all patients was 20.3 months [50].

Another study by Sang et al., which was a phase II clinical trial of co-administration of CD-19- and CD-20- targeted CAR-T cells in 21 adult patients with R/R DLBCL, showed the objective response rate at 3 months of 81% (17/21) and CRR of 52.4% (11/21). The 6-month sustained ORR and CR rates were 46.2% (6 patients) and 40.0% (4 patients), respectively. The median PFS, OS, and DOR were 5.0 months, 8.1 months and 6.8 months, respectively. PFS and DOR were not as encouraging as CR and ORR, probably because of immune escaping, lacking patient-originated memory CAR-T cells, or failure of continued CAR-T cell expansion. However, in this study, three out of four patients with MYC/BCL-2 double expression, who were resistant to rituximab-based immunochemotherapy, achieved CR. Moreover, one patient of CD5-positive DLBCL with testicular involvement also achieved CR, suggesting that anti-CD19 and anti-CD20 CAR-T cells are capable of passing the blood–testis barrier [51].

##### CD19/CD22

CD22, a member of the sialic acid-binding immunoglobulin-like lectin (Siglec) family, is an inhibitory co-receptor of B-cell receptor that is exclusively expressed on B-cells. CD22 is highly expressed on B-cell lymphomas and leukemias, so it has become a therapeutic target of cell therapy [52]. Clinical trials of CD19/CD22 bispecific CAR-T cell therapy have manifested encouraging efficacy in B-cell malignancies in both adults and children.

Recently, a phase I/II clinical trial studied the effectiveness of a humanized and murinized version of CAR19/22 T-cell cocktail therapy for R/R aggressive B-NHL in both children and adults (4–75 years old). The study revealed that CAR19/22 T-cell therapy is safe and effective for R/R B-cell lymphoma and that patients treated with a humanized CAR-T exhibited better efficacy, compared to patients treated with a murinized CAR-T, even among TP53 mutation-positive patients. In the humanized CAR-T group, an objective response by day 28 post-infusion was achieved in 91.7% patients (11/12) and CR—in 91.7% of patients (11/12). In the murinized CAR-T group, ORR and CR were 92.9% (13/14) and 42.9% (6/14), respectively. Moreover, 75% of the patients (9/12) in the humanized group maintained CR at month 3 following infusion, compared to 35.7% of the patients (5/14) in the murinized group. Also PFS was more favorable in the humanized CAR-T cell therapy. Interestingly, CAR19 or CAR22 T-cells were still detected in the peripheral blood, cerebrospinal fluid and hydrothorax months to one year later after receiving humanized CAR T-cells, which may have the potential to mediate long-term disease remission [53].

Shalabi et al. in their phase I clinical trial also aimed to evaluate the effectiveness of CD19/CD22 CAR-T cell therapy in children and young adults (5–34 years old) with B-cell malignancies. A total of 20/21 children were diagnosed with B-cell ALL and only 1 patient with NHL (refractory BL). Unfortunately, the patient with NHL was not included in the analysis, because she had subsequently died from progressive disease 5-months post-CAR-T cell therapy [54].

Phase II clinical trial by Qu et al. tested the efficiency of tandem CD-19/CD22 dual-targeted CAR-T cell therapy in 33 adult patients with R/R DLBCL pretreated with DFC lymphodepletion chemotherapy (decitabine, fludarabine plus cyclophosphamide). Within 1 month and 3 months after CAR-T infusions, 90.6% (29/32) and 93.8% (30/32) of the patients achieved an objective response, with 28.1% (9/32) and 40.6% (13/32) achieving CR. With a median follow-up of 10.9 months, the best ORR and CR rates were 90.9% and 63.6%, respectively. The median PFS was 10.2 months and the OS was undefined. The 2-year OS and PFS rates were 54.3% and 47.2%, respectively. Statistical analysis revealed that achieving CR was an independent prognostic factor associated with favorable PFS and OS. Unfortunately, there were three cases who relapsed after CAR-T cell therapy. They underwent allogeneic hematopoietic stem cell transplantation (allo-HSCT) and two of them (2/3) achieved CR and one (1/3) achieved PR (but died of severe GVHD). In this study, the patients showed high efficacy, good survival and reversible toxicities, highlighting that DFC chemotherapy may be an effective and safe lymphodepletion regimen before CAR-T therapy in high-risk DLBCL patients and for those who relapsed after CAR-T therapy allo-HSCT may be a feasible salvage approach. Additionally, in this trial, 16 patients not achieving CR within three months received maintenance therapies, 6 patients with lenalidomide, 2 patients with BTKi, 5 with lenalidomide and BTKi, and 3 with radiotherapy. Finally, 5/6 patients who had been given lenalidomide achieved CR and were alive, and 5/5 patients who had got lenalidomide and BTKi achieved CR, where 4 patients were alive and one died of PD after unauthorized withdrawal of drugs. However, two patients who had received BTKi and three patients exposed to radiotherapy were irresponsive and died of PD. These data support the use of lenalidomide-containing regimen to enhance CAR-T cell efficacy for patients not achieving CR after CAR-T therapy in DLBCL [55].

Spiegel et al. in their phase I clinical trial evaluated the efficiency of CD19/CD22 CAR-T cell therapy in adults with R/R B-cell ALL and LBCL. In the LBCL group, the best ORR at any time point was 62% (13/21 patients) and the CR—29% (6/21). With a median follow-up of 10 months, the median OS was 22.5 months, but it could change with a longer follow-up. The median PFS was 3.2 months. Interestingly, the researchers assessed a change in lymphoma burden over time by cell-free circulating tumor DNA (ctDNA). Four patients with ongoing clinical response had no detectable ctDNA at the time of the last assessment. Among 12 patients with disease progression, they observed an initial reduction in ctDNA that endured 14–21 days post-infusion, with 9 patients demonstrating a rise in ctDNA at or before clinical progression. These findings suggested that progressive disease after CD19/CD22 CAR-T in LBCL may be associated with a robust early response followed by early acquired resistance [56].

Another study by Cao et al. aimed to investigate the safety and efficacy of sequential infusion of CD19/22 CAR-T cells following auto-HSCT in adult patients with R/R aggressive B-NHL after salvage chemotherapy. The study enrolled 23.8% of patients (10/42) with PR and 76.2% (32/42) with SD or PD before auto-HSCT after salvage treatment. At 3 months post-treatment, 90.5% of the patients (38/42) experienced a response, including 81% (34/42) with a CR and 9.5% (4/42) with a PR as assessed by PET-CT. At the data cutoff date, 31/34 patients who had a CR at 3 months maintained their durable response. All four patients who had a PR at 3 months improved to CR at a median time of 3 months without additional antilymphoma therapy. At a median follow-up of 24.3 months, PFS was 85.7% at 1 year and 83.3% at 2 years and the 2-year OS rate was 83.3%. Moreover, no patients were found to be CD19- and CD22-negative at the time of progression, and 97.1% and 68.6% of the patients with ongoing CR had consistently detectable levels of CD19 and CD22 CAR transgene, respectively, at 3 months. The high durable CR rates and favorable safety profiles obtained in this study, support the strong potential of auto-HSCT plus CD19/CD22 CAR T cell therapy for patients with failed salvage therapy of NHL [57].

Study by Wu et al., evaluated the outcomes of CD19/CD22 CAR-T cell immunotherapy, both alone and in combination with auto-HSCT in patients (aged 17–70) with R/R BL. A total of 89.3% of the patients (25/28) had advanced-stage disease and 71.4% (20/28)—a high-risk subtype with genetic abnormalities including *TP53* mutations, *ID3* mutations, *DDX3X* mutations and *MYC* mutations. A total of 67.9% of the patients (19/28) eventually responded to CAR-T cell therapy, with 57% of the patients (16/28) achieving CR and 11% (3/28) achieving PR. The ORR was 37.9% (19/28) and CR—57.1% (16/28). None of the participants who achieved CR experienced recurrence during long-term follow-up. Moreover, 62.5% of the patients (10/16) with CR status stayed in durable remission for >18 months. When comparing the outcomes between groups, better results were achieved in the group treated with CD19/CD22 CAR-T cell and auto-HSCT. The ORR and CR in the group treated with combination of CAR-T and auto-HSCT were 92.3% (12/13) and 84.6% (11/13), compared with ORR and CR in the group treated with CAR-T cell only—46.7% (7/15) and 33.3% (5/15), respectively. After a median follow-up of 12.5 month duration, 16 patients survived. Both the estimated 1-year progression-free and OS rates were 55.6%, but the estimated 1-year PFS and OS rates of the patients were higher in the group treated with combination of CAR-T and auto-HSCT, compared to CAR-T cell therapy alone—83.3% vs. 33.3%. In conclusion, CD19/CD22 CAR-T cell infusions combined with auto-HSCT appeared to lead to more favorable long-term outcomes in adult patients with R/R BL and may be used as salvage therapy [58].

Zhou et al. also reported sequential CAR19/22 T cell therapy response rates in six young adult patients with refractory BL with high-risk genetic abnormalities (including *ID3* mutation, *TP53* mutation, *DDX3X* mutation and *MYC* mutation). A total of 3/6 patients (50%) achieved an objective response, including 2 PRs (33.3%) and 1 CR (16.7%). One CR patient received allo-HSCT and was in remission at 37 months post-infusion. The other three patients who achieved CR were enrolled into another clinical trial that consisted of sequential infusion of CAR22/19 T cells following autologous stem cell transplantation (auto-HSCT). Although both CAR-T cells expanded in all patients, both CAR22 and CAR19 copies in effective cases were higher than in ineffective cases ending with PD [59].

Zeng et al. examined the efficiency of sequential infusion of anti-CD22 and anti-CD19 CAR-T cells in 14 adult patients with R/R aggressive B-NHL involving gastrointestinal tract. The study revealed the ORR rate of 76.9% (10/13), with 53.8% (7/13) CRs and 23.1% (3/13) PRs. The median duration of PFS was 6.2 months. The OS and PFS rates at 6 months after infusion were 71.4% (10/14) and 50.0% (7/14), respectively. Moreover, the patients with CR had significantly superior OS and PFS than those with stable disease. A total of 85.7% (6/7) of the patients who had achieved CR maintained durable remission by the study cut-off at 21.5 months post-infusion. Unfortunately, a loss of both CD19 and CD22 antigens targeted by CAR-T cells was observed in two patients, who developed progressive disease. Notably, all of the patients, regardless of CR, PD or the loss of CAR transgene copy, had B-cell aplasia. Although B-cell aplasia was present in most patients before CAR-T therapy due to treatment history of CD20 antibodies, patients monitored for more than 6 months (*n* = 6) had persistent B-cell aplasia, and the longest duration reached 21.5 months [60].

#### 2.3.3. Sequential CD19/CD20/CD22-Targeted CAR-T Cells

In recent years, most of the research concerned with the use of CAR-T cell therapy in children with NHL was based on sequential administration of different B-cell antigen-targeted CAR T-cells. This strategy is designed to prevent tumor antigen escape and maintain CAR T-cell persistence.

Zhang et al. aimed to examine the efficacy and safety of multitargeted CAR T-cell therapy for pediatric patients with mature B-cell lymphomas. The study enrolled five patients with R/R BL aged 6 to 10 years. Previously they had all failed to show response to multiple courses of intensive chemotherapy and anti-CD20 mAb treatment. A total of 3/5 patients were at stage IV and 2/5 were at stage III, according to the St. Jude staging system. A total of 3/5 children achieved CR after one round of CD19 CAR-T cell therapy. One patient showed a transient response to CD19 CAR-T cell therapy, probably due to CD19 CAR-T cells not expanding adequately, so he received CD22 CAR-T cell therapy as soon as the tumor regrowth was detected. Then, he finally achieved CR. Another patient had no response to the first round of CD19 CAR-T treatment, probably due to delayed expansion of CD19 CAR-T cells, so he also received CD22 CAR-T cells, which seemed to expand adequately. On day 45 post-infusion PET-CT revealed a disappearance of the original tumor, but also an emergence of a new mass, so he was given CD20 CAR-T cells. On day 64 PET-CT confirmed CR. To conclude, CR was observed in all patients between 37 and 77 days post-infusion. The ORR of this multiple CAR-T infusion approach was 100%, including patients who did not respond or progressed after prior CAR-T products, with a median follow-up of 331 days, ranging from 149 to 428 days. All patients remained in CR at the end of the study. The toxicities of the treatment were tolerable. All patients showed myelosuppression, including anemia, thrombocytopenia, and neutropenia, likely attributed to lymphodepleting chemotherapy, which required blood transfusion. All patients showed CRS of different grades, but they fully recovered after active symptomatic and supportive treatment, including an application of corticosteroids in CRS grade 3. This study revealed that targeting different B-cell markers sequentially in CAR-T cell therapy is effective for treating R/R BL in children [61].

Recently, the same group of researchers conducted a clinical trial, which is the largest published so far, on the use of CAR-T cell therapy in children with NHLs. The study enrolled 23 children with R/R BL. At study entry, 13% (3/23) of the children were in stage II of the disease, 43.5% (10/23) in stage III, and 43.5% (10/23) in stage IV. Overall, 91% (21/23) of the patients had risk factors for treatment failure due to R/R disease within 6 months of diagnosis, an elevated LDH ≥2 times the upper limit of normal at diagnosis, or a failure in the bone marrow. Additionally, 39.1% (9/23) of them had bulky disease and 43.5% (10/23) had CNS involvement. Firstly, all patients received CD19 CAR-T cells and those who did not achieve an ongoing CR underwent one or more sequential infusions of CAR T-cell therapy that targeted CD22, followed by CD20. After the first infusion of CD19 CAR-T cells 65.2% (15/23) of the patients achieved CR and 26.1% (6/23) achieved PR. A total of 60% (9/15) of the patients who achieved CR maintained an ongoing CR and 40% (6/15) developed a relapsed disease within 1.5 to 6 months post-infusion. A total of 13 patients received a second cycle of infusion with CD22 CAR-T cells. Among them, 38.5% (5/13) achieved an ongoing CR, 23% (3/13) achieved CR, 15.4% (2/13) attained a PR, 7.7% (1/13) developed PD after a transient PR and 15.4% (2/13) died of rapid intracerebral mass (ICM) progression. Then, six patients received the third cycle of infusion with CD20 CAR-T cells. Among them, 50% (3/6) achieved an ongoing CR, 16.7% (1/6) remained in CR for 5 months and then relapsed, 16.7% (1/6) died because of CNS disease relapse. A total of one patient who showed a PR received the fourth cycle of CD19 CAR T-cell infusion and ultimately attained an ongoing CR. Overall, the median time from the last infusion to the cutoff date was 17 months. A total of 91.3% (21/23) of the patients achieved a CR within 3 months after the last infusion and 78.3% (18/23) remained in CR to the cutoff date. Durable responses may be partly attributable to retained functional CAR T-cell persistence, as indicated by 51% of the patients estimated to have B-cell aplasia at 12 months after the first infusion. The estimated 18-month CR rate was 78%, OS rate—83% and PFS—78%, with 78% in the patients with bulky disease and 60% in the patients with CNS involvement. A total of 50% of the patients were estimated to have CAR T cells detectable in peripheral blood at 6 months after first infusion and levels of CAR transgenes were still detectable for up to 35 months after treatment. During the first CD19 CAR T-cell infusion, grade ≥3 CRS occurred in 34.8% and neurotoxicity occurred in 21.7% of all the patients. During subsequent infusions, there were only a few incidences of grade >2 CRS and neurotoxicity. All adverse events were reversible and the severity of neurotoxicity was not significantly different between patients with CNS involvement and without [62].

Du and Zhang described a case of an 8-year-old boy with R/R BL treated with sequentially targeting CD19, CD22, and CD20 with CAR-T cells. He had no discernible response to anti-CD19 CAR-T treatment and exhibited PD, so then he received CD22 CAR-T cells and underwent a PR. Unfortunately, a relapse rapidly occurred, so he was given CD20 CAR-T cells and finally went into CR, which was revealed in PET-CT 65 days post-infusion. By the end of follow-up he achieved 16-month event-free survival (EFS). The toxicities of the treatment were tolerable. The boy experienced mild CRS (grade 1) during administration of the CD19 and CD20 CAR-T cells and CRS grade 3 during the CD22 CAR-T therapy. However, at approximately 6-month follow-up after the end of treatment, the patient’s red blood cells, white blood cells, and platelets based on his routine blood panel, had returned to normal, and his other organs also functioned properly. CAR-T cell therapy targeting multitumor antigens showed a satisfactory effect in the described case [63].

Zhang et al. also conducted a clinical trial assessing the efficiency of sequential CAR-T in the treatment of R/R B-NHL in 17 pediatric patients. The study enrolled children with R/R BL (13/17), DLBCL (2/17) and B-lymphoblastic lymphoma (2/17) aged 4.5 to 18 years. By St. Jude staging, 46.7% (9/17) of the patients were in stage III and 53.3% (8/17) in stage IV. A total of 17.6% of the children had CNS involvement (3/17) and 41.2% (7/17) had bone marrow involvement. Firstly, CD19 CAR-T cells were administered. After the first infusion, the overall CRR was 41.7% (7/17). Among the 10 patients who did not achieve CR, 2 patients achieved PR with ongoing response, 1 patient died of severe CRS and 1 because of disease progression The other six continued to receive the second course of CAR-T therapy targeting CD20 or CD22, and three of them achieved CR. Thus, the overall CRR increased to 58.8% (10/17). The three patients who still did not achieve CR continued to receive the third course of CAR-T therapy targeting CD20 or CD22, and two of them finally achieved CR. Eventually, with a median follow-up of 6.2 months, the ORR of sequential CAR-T therapy was 94.1% (16/17) and the overall CRR was 70.6% (12/17). On day 30 post-infusion, 47.1% (8/17) of the children had mild CRS (grade I) and 52.9% (9/17) severe CRS (grade III or IV). Neurotoxicity was observed in 41.2% (7/17). A total of 94.1% (16/17) of the patients with CRS and neurotoxicities recovered fully after glucocorticoid use and symptomatic treatment [64].

In their next study, they confirmed that short interval sequential infusions of CD19/CD22/CD20 CAR-T cells could enhance expansion of prior CAR-T cells with stronger tumor-killing capacity. They proved that in in vitro and in vivo trials and then administered short interval sequential infusions to two patients with BL and DLBCL (from a prior study) who had disease progression after primary CAR-T cell therapy. The first patient, a 15-year-old diagnosed with BL, underwent a PET-CT examination on day 100 after infusion, which revealed a residual tumor mass which did not display the typical malignant tumor-like hypermetabolic lesions and was subsequently resected. The tissue mass was examined by FCM and it showed no viable tumor cells in the surgical specimen. The second patient, a 15-year-old girl with DLBCL with *TP53* mutation, underwent PET-CT scan on day 130 post-infusion, which revealed that the patient had achieved CR. This study revealed that sequential CAR-T cell infusions may induce co-expansion of different CAR-T cells when residual prior CAR-T cells still remain detectable in peripheral blood, leading to a prolonged duration of peak expansion of CAR-T cells with enhanced antitumor effects [65].

#### 2.3.4. General Relevance of Clinical Trials on the Effectiveness and Safety of CAR-T Cell Therapy to the Pediatric Population

The above-mentioned studies involving pediatric patients indicate that CAR-T cell therapy may be effective and safe among children or adolescents with NHL. Also its treatment results are similar to those in adults. This conclusion applies to CD19-targeted CAR-T cell therapies (only tisa-cel and KTE-X19), bispecific targeted CAR-T cell therapies (only CD19/CD22), and sequential CD19/CD20/CD22-targeted CAR-T cell therapies. Unfortunately, no pediatric clinical trial results are available for other types of targeted CAR-T cell therapies. Therefore, their effectiveness and safety in children and adolescents are currently unknown. However, recruiting clinical trials have great potential.

### 2.4. Recruiting Clinical Trials

The findings of numerous clinical trials conducted on adults, showing the effectiveness of CAR-T cell therapy in NHL treatment, resulted in the commencement of clinical trials on the pediatric population. Currently, 44 recruiting phase 1, 2 or 3 clinical trials are being conducted, testing the possibility, effectiveness and safety of the use of, among others, anti-CD19, -CD20, -CD22, -CD30, -CD5, and -CD7 CAR-T cells in children with both B-cell and T-cell NHL. For more information on currently recruiting clinical trials, see Table 3. It presents studies conducted among children, adolescents and adults, because it was not possible to distinguish only subgroups of children and adolescents, due to broad age classification criteria.

## 3. The Challenges of CAR-T Cell Therapy

### 3.1. Aftermath of CAR-T Cell Therapy

Unfortunately, the success of CAR-T cell therapy comes with a price, and occurrence of adverse effects (AEs) is inevitable. CRS and immune effector cell-associated neurotoxicity syndrome (ICANS; previously also referred to as CAR-T-cell-related encephalopathy syndrome or CRES) are in the lead of CAR-T cell therapy AEs and thus remain a crucial element of the aftermath. Likewise, cytopenias, infections or tumor lysis syndrome (TLS) continue to be troublesome [109]. Management of CAR-T cell therapy AEs grows in its significance as it is breaking through into the earlier stages of treatment.

#### 3.1.1. Cytokine Release Syndrome

Cytokine release syndrome is the most frequently occurring adverse event of CAR-T cell therapy. The prevalence of CRS is not necessarily affected by cancer type, co-stimulatory domains or trial phase [110]. It has been speculated that the development of CRS may correlate with successfulness of CAR-T therapy, but it requires yet further exploration. Its exact mechanism is not yet entirely understood. Nonetheless, it is related to the release of inflammatory cytokines during CAR-T cell administration [111].

As it is a systemic inflammatory reaction, fever is the dominant and the earliest symptom of CRS, and it can reach more than 40.5 degrees Celsius in a few days [112]. Other symptoms include fatigue, myalgia, arthralgia, rigors or anorexia. Furthermore, these alterations may be followed by tachycardia, hypotension in necessity of vasopressors, tachypnea and hypoxia, or neurological changes [113]. Ultimately, coagulopathy and capillary leak as well as respiratory distress, shock or even multiple organ failure may develop, and the patient will urgently require hospitalization at the intensive care unit (ICU). Luckily, CRS-related deaths have hardly been reported [114].

CRS appears in the course of 1 to 14 days and its onset tends to settle after approximately 14 to 21 days [112,115]. Risk factors for severe CRS include, inter alia, a high infusional dose of CAR-T cells or early cytokine elevations, whereas prevention strategies hinge upon pretreatment cytoreduction or prophylactic cytokine therapy [112].

Since the cytokine profile correlates with the clinical onset of CRS, the diagnostic process of severe CRS necessitates estimation of IL-6, IL-10, with some researchers proposing to also verify interferon-γ (IFN-ɣ), tumor necrosis factor-alpha (TNF-ɑ), and/or IL-5 levels [116,117]. Serum levels of C-reactive protein (CRP) and ferritin are mandatory to perform. Markers such as complete blood count (CBC), coagulation profile, renal and liver function test, and LDH also help determine prognosis for CRS. A complete evaluation ought to comprise clinical examination after a lapse of 21 days of CAR-T cell administration.

CRS can either be self-limited (with the possible support of antipyretics and intravenous fluids) or inevitably call for intervention with anticytokine-directed therapy. The most standard management of CRS, approved by the US FDA consists of tocilizumab, a humanized mAb against IL-6 receptor. Tocilizumab abates CRS promptly, and, even more importantly, it does not interfere with therapeutic effects of CAR-T [118]. Interestingly, tocilizumab with a subsequent anti-CD19 CAR-T cell administration was reported to decrease the prevalence and severity of CRS used in prophylaxis, but those trials did not involve children or adolescents [119]. In cases of tocilizumab insufficiency, corticosteroids are added to help control CRS, but in high doses, in contrast to tocilizumab, they may damage the desirable effect on CAR-Ts. In the study by Jiang et al., disseminated intravascular coagulation (DIC) is recognized as another CAR-T cell-related event caused by CRS [120]. It is implied that corticosteroids as well as immunosuppressive agents may preclude coagulation in CRS patients, including children and adolescents. In the research by Jess et al., it is reported that CRS in CD22 CAR-T therapy is relatively stable in its prevalence and severity [121].

#### 3.1.2. Immune Effector Cell-Associated Neurotoxicity Syndrome

Immune effector cell-associated neurotoxicity syndrome (ICANS) occurs in a significant number of patients after CAR-T cell administration. However, in comparison to CRS, ICANS occurrence is delayed. Its pathophysiology remains vague, but existing theories claim that it may correspond to CNS T-cell trafficking, increased vascular permeability, and endothelial disruption in the blood–brain barrier (BBB) [116].

Clinical signs and symptoms of ICANS are various and wide in their range: from mild headaches through cerebral edema and acute encephalopathy to aphasia, focal deficits, confusion, delirium, seizures or even visual hallucinations [122,123]. Dysgraphia has been noted as an early indicator of ICANS. ICANS tends to develop after CRS onset.

Thus, the presence of CRS is one of the risk factors for ICANS amongst others, such as pre-existing neurologic dysfunction, elevated LDH and ferritin in laboratory tests, and thrombocytopenia.

Since the immune effector cell-associated encephalopathy (ICE) used to grade ICANS in adults is unadjusted for pediatric patients, Traube et al. adapted the Cornell Assessment of Pediatric Delirium (CAPD) and it is now a screening tool for recognizing delirium in children [124,125]. CAPD is the most valid in patients under 12 years old. The guidelines suggest assessment at least twice per day [124,126]. The ICANS grading in pediatric patients combines the ICE (for children ≥12 years olds) and CAPD (for children <12 years old) into overall scale for neurological examination, which includes orientation, naming, following commands, writing and attention (ICE), as well as eye contact with a caregiver, purposeful actions or communicating needs and wants (CAPD).

Evaluation of ICANS also comprises laboratory tests similar to CRS along with an cerebrospinal fluid analysis, neuroimaging and electroencephalography (EEG). All these components aid in estimating the degree of ICANS damage.

ICANS management strategies consist primarily in minimization of inflammatory response. Mild ICANS patients are offered intravenous hydration. Corticosteroids (dexamethasone or methylprednisolone) are recommended for ICANS grade 2 or higher. Due to its poor BBB penetration, tocilizumab has no contribution in ICANS management unless there is a concurrent CRS [126].

Another IL-6 antagonist mAb, siltuximab, is considered to take a meaningful part in ICANS management as it wards off IL-6 transfer from BBB [122]. Moreover, it is speculated that in ICANS antiepileptic drugs are used to prevent not only seizures, but also severe neurotoxicity [127]. However, these hypotheses remain unclear and need further exploration. A finding by Jess et al. provides an opportunity for reducing ICANS incidence and severity by replacing CD19 CAR-T with CD22 as they differed in their expression in CNS [121].

#### 3.1.3. Other Neurotoxicities

An interesting finding has been presented concerning another form of CAR-T-related neurotoxicity. Ruark et al. published an article on patient-reported neuropsychiatric outcomes [128]. Nearly half of the adult patients observed at least one clinically significant negative neuropsychiatric outcome, such as cognitive difficulty, anxiety or depression. As the results suggest, probable risk factors for long-term neuropsychiatric problems are younger age, pre-CAR-T anxiety or depression, and acute neurotoxicity. However, further research is essential to confirm those outcomes. Obviously, it still remains to be determined if those findings apply to children and adolescents.

#### 3.1.4. Infections

Infections are yet another challenge related to CAR-T therapy. The therapy negatively influences a patient’s immune system, principally by causing depletion of B-lymphocytes, with low immunoglobulin rate and prolonged leukopenia. The underlying disease, for which CAR-T cell therapy was used in the first place, high doses of CAR-T cells and past chemotherapy incidents predispose to severe infections. Equally, the presence of the CRS or/and ICANS increase the risk of infections as their management requires even further intervention in the functioning of the patient’s immune system.

It is recommended that pediatric patients previously treated with HSCT be monitored for adenovirus, cytomegalovirus (CMV), Epstein–Barr virus (EBV) and human herpesvirus 6 (HHV6). Acyclovir is used for prophylaxis in herpes simplex virus (HSV)-positive children. Interestingly, anti-CD19 CAR-T cell therapy does not interfere with the patients’ concomitant hepatitis B virus (HBV) infection and thus, it could be safely used, as presented in a trial on adult patients [129].

Routine prophylaxis is not implied for bacterial infections. Fungal diseases are most likely connected to CRS or HSCT. Antifungal prophylaxis should be considered, especially in patients at high risk, e.g., those with prolonged steroid exposure or prolonged and/or severe leukopenia. Pneumocystis jirovecii (PJP) prevention is recommended for pediatric patients [130,131].

#### 3.1.5. Cytopenia

Cytopenias occur fairly commonly in patients after CAR-T cell therapy, of which neutropenia is the most widely spread. Factors such as age, gender or total previous treatment seem to play a role in the prevalence of post-CAR-T therapy cytopenias [132]. Early cytopenia generally manifests itself within 28 days after CAR-T cell administration [133]. In its nature, CAR-T-related cytopenia may be biphasic, or even triphasic [134]. The onset of early cytopenia is associated with the intensity of bridging chemotherapy or pre-administration lymphodepletion chemotherapy, to name a few. Patients with severe CRS or secondary hemophagocytic lymphohistiocytosis (HLH) are in danger of developing stem cell exhaustion as they have a significantly increased level of IFN-ɣ, which causes a suppression of stem cell homeostasis [135].

If cytopenia lasts for longer than 3 months after the infusion, it is classified as prolonged cytopenia. Its mechanism is persistently vague. However, it is suggested that factors such as severe CRS, allo-HSCT within 1 year or baseline cytopenia increase risks of late cytopenia [136,137].

Prolonged cytopenia might certainly expose a fragile patient to opportunistic infections. In order to prevent those, it is recommended to administer the patient’s granulocyte colony-stimulating factor (G-CSF) two weeks after the infusion [138].

#### 3.1.6. B-Cell Aplasia and Hypogammaglobulinemia

On-target, off-tumor effect of anti-CD19 and anti-CD22 CAR-T cells on normal B-cells generates B-cell aplasia and subsequent hypogammaglobulinemia. BCA might progress into either a manageable depletion and helpful tool to estimate the effectiveness of the applied treatment, or a persistent form leading to hypogammaglobulinemia, eventually resulting in infections.

This AE is to be managed with a use of periodic intravenous immunoglobulin (IVIG) supplementation [124,126,138]. In pediatric patients, IVIG routine replacement is recommended as a standard treatment [126,133,139].

#### 3.1.7. Hemophagocytic Lymphohistiocytosis

This fulminant hyperinflammatory syndrome is likely to arise in patients suffering from serious infection, autoimmune disease or malignancy. Despite its relative rarity (in data gathered in adult patients), this multiorgan failure grows in incidence and, more importantly, has an inglorious high rate of mortality [132,140]. CRS tendency to progress into or overlap with HLH makes lymphohistiocytosis more challenging in its management. HLH manifests itself mainly with fever, jaundice, organomegaly, as well as gastrointestinal and pulmonary problems [140,141]. Although diagnostic criteria for carHLH were suggested, such as peak serum ferritin level and development of organ failure (hepatic, renal, pulmonary), the scale is for now recommended for adults only and needs adjustment for pediatric patients [142]. Interestingly, in contrast to ICANS, CD22 CAR-T therapy in children and adolescents elevates the incidence of HLH-like toxicities [121].

#### 3.1.8. Tumor Lysis Syndrome

As a common consequence of anticancer treatment, malignant cells disintegrate. This phenomenon is known as tumor lysis syndrome (TLS). This process is followed by metabolic abnormalities such as hyperkalemia, hyperuricemia, hyperphosphatemia and hypocalcemia. Those may cause arrhythmias and renal failure. Crucial elements of TLS management comprise hyperhydration and allopurinol or rasburicase [143,144,145].

#### 3.1.9. Anaphylaxis and Immunogenicity

Most CAR-T cells contain the addition of non-human elements rendering a risk for allergic reactions. Anaphylaxis is said to happen occasionally but more so in children, as it has been described mostly in cases of repeated CAR-T cell administration [146]. Fully humanized CARs are under clinical trials in order to reduce immunogenicity [147,148]

There continues a lack of data presenting CAR-T cell therapy effect on adult as well as pediatric patients’ experience, or the duration and quality of life and, hence, it is undoubtedly a vast field for future exploration [110].

### 3.2. Limitations of CAR-T Cell Therapy

AEs in CAR-T cell therapy are not the only challenge which present-day medicine is compelled to face. Others include the limitations of CAR-T cell therapy, such as patient selection, resistance to treatment or the tumor microenvironment (TME). Factors like race, ethnicity, gut microbiome and tumor burden (TB) influence response to CAR-T therapy as well.

#### 3.2.1. Patient Selection

As trivial as it may seem, each patient responds to treatment in an individual way. In patient selection, age, fitness, previous therapies, concurrent diseases and organ function, along with practical aspects, such as logistics of administration, must be taken into consideration as they affect the patient’s reaction to CAR-T therapy [149,150].

Moreover, it is important to have in mind that tisa-cel, axi-cel and liso-cel are different in terms of their composition, manufacturing or toxicity profile and thus, it is essential to choose the most suitable product for the patient, taking into account such specifics as eligibility and timing of CAR-T cell administration [151].

Additionally, bridging therapies and lymphodepleting grow in their importance as elements of CAR-T cell treatment [151]. Bridging therapy may consist of standard chemotherapy or immunotherapy and/or radiotherapy.

#### 3.2.2. Resistance

A multitude of components may contribute to the development of a patient’s resistance to CAR-T cell therapy. Obstacles might occur on different stages of treatment.

##### Obstacle 1: Achieving CAR-T

Access. Since FDA approval of axi-cel in 2017, the situation has improved as CAR-T cell therapy has become internationally accessible [152]. Despite a worldwide range, there is still a question of local accessibility. Treatment centers must have both the capability and the capacity to comply with FDA protocols of manufacturing and administration of CAR-T to the patient, and their number is limited [153]. Another question concerns the high expense and its coverage [154].

Production. It is crucial that CAR-T cells are manufactured and administered successfully. In order to achieve this, the patient must possess a sufficient number of T-lymphocytes for following collection and harvesting. Successful manufacturing applies to both quantity and quality of T-cells [155]. CAR component design contributes to the characteristics of CAR-T cells and their consecutive in vivo performance, such as expansion and persistence. Alternatively, a CAR-T cell product can be generated from allogeneic cells. Such a manner of manufacturing CAR-T cells is considered a method for overcoming quantitative or qualitative insufficiency of autologous CAR-T cells. In order to produce universal off-the-shelf CAR-T cells, T-lymphocytes are collected from healthy donors with no prior exposure to immunosuppressive chemotherapy and genetically modified by either the CRISPR/Cas9 or TALEN system [156,157]. Allogeneic CAR-T cell therapy is still in development in clinical trials, involving both adult and pediatric patients.

Infusion. Timing of administration has a role in CAR-T cell therapy success. A patient must receive an infusion of CAR-T cells prior to the development of a progressive disease or disease-connected complications. Otherwise, therapy is likely to fail. Future strategies hinge upon attempts to reduce production time in order to increase the number of beneficiaries.

Activation and expansion. Both of these factors are substantial for CAR-T cell therapy. It has been exposed that small doses (0.2–5.0 × 10^6^ transduced CAR T cells per kg or a total of 0.1–2.5 × 10^8^ transduced CAR T cells per infusion) suffice to induce improvement. Nonetheless, factors such as quality of T-cells, patient’s individual burdens and lymphodepletion affect post-infusion cell behavior.

##### Obstacle 2: Relapse

Relapse subsequent to either anti-CD19 or anti-CD22 therapy is an obvious challenge. It may develop generally in two patterns: early antigen-positive relapse or later relapse with antigen loss (both best explored in patients with pre-B cell ALL) [133,149].

Antigen-positive relapse. Relapse with antigen-positive disease creates a promising opportunity for re-treatment with CAR T cells. In a study by Turtle et al., poor outcomes subsequent to re-infusion in patients with B-NHL were reported [158]. However, fludarabine combined with cyclophosphamide used to intensify lymphodepletion enhanced the re-infusion response, along with initial CAR-T cell expansion and persistence. Other approaches for optimizing re-infusion involve the use of an alternative CAR construct or CAR targeting a different antigen, e.g., CD22.

Antigen loss or modulation. Target antigen modulation mechanisms consist in a loss of CD19 expression. One way for this to happen is through alternative splicing, which generates CD19 isoforms with disruption of the target epitope and/or reduced cell surface expression [159]. Another mechanism is interrupting CD19 transport to the cell surface [160]. Interestingly, total antigen loss may not be mandatory for resistance to an originally successful CAR-T cell therapy to develop. As it turns out, a simple decrease in antigen expression suffices to achieve that. Lineage switching is yet another mechanism for escape from CAR-T cells. Regarding the above-mentioned inclinations, CAR constructs combining multiantigen targeting are being developed to address tumor heterogeneity and, thereby, decrease the risk of relapse.

One opportunity for vanquishing such a phenomenon is manufacturing CAR-T cells simultaneously targeted at multiple antigens by means of either bicistronic CARs or tandem CARs (this particular research described it in multiple myeloma) [161]. Alternatively, a sequential infusion of two products targeting different antigens may be introduced [162].

Armored CAR-T cells are to boost activation of antitumor cells as well as increase their recruitment toward antigen-independent cancer killing through secretion of cytokines, e.g., IL-12, IL-18, and expression of ligands, such as CD40L [163]. Both IL-12 and IL-18 have an antitumor effect: the first by increasing the effectiveness of TRUCKs, and the latter by intensifying endogenous immunity in the TME [164]. Another strategy engages creating CAR-T cells expressing CD40L or other costimulatory ligands, such as CD80 or 4-1BBL [165].

#### 3.2.3. The Tumor Microenvironment

The TME is an intricate complex, which comprises cells evolving with malignant cells and aiding them in their malignant transformation. In addition, the TME consists of the tumor vasculature, lymphatics and the immune system, along with fibroblasts and adipocytes. All types of cells and molecules present in the TME may take part in tumor development and progression, including promotion of angiogenesis, inducement of drug resistance or immune system suppression [166].

As proved in the studies, composition of the TME contributes significantly to the prognosis, for example presence of T-cells and/or NK cells in the TME as well as high expression of programmed death (PD) ligands in malignant cells are predictors for good results [167,168]. Other findings were presented by Yan et al.; patients with CR displayed low levels of immunosuppressive proteins (CCL2, CXC8, CXCL12, CCL3, -4, -5) affecting the TME, whereas in the group with partial remission (PR) immunosuppressive factors such as CXCL9 were overexpressed [169]. Additionally, immunosuppressive cytokines, e.g., IL-10, or tumor-associated fibroblasts (STC1, etc.) were increased in the PR patients, in comparison to CR ones. Consequently, it is speculated that the TME is likely to conceive a milieu to thwart CAR T-cell antitumor activity.

Therefore, augmentation of PD-L1 and PD-L2 expression or increase in the number of recruited and activated local T-cells appear to be a promising method of surpassing the TME limitation in CAR-T cell therapy. Additionally, to advance CAR-T cell therapy in terms of the TME, one requires optimizing the dose and timing, or introducing CAR-T therapy earlier in the disease course, to name a few potential remedies [170]. Nevertheless, there is not enough investigation data to safely determine strategies to overcome the TME barrier in children and adolescents.

Moreover, a study by Faramand et al. draws attention to the TME in its proinflammatory nature and the way it influences CAR-T-related toxicities, CRS and ICANS, in adult patients with LBCL [171]. To prevent the unfavorable impact of the TME, it is speculated that modulating it towards T-cell infiltration phenotype, accompanied by curbing systemic inflammation might come to be a potential strategy to lessen toxicity.

#### 3.2.4. Gut Microbiome

A study by Hu et al. concentrates on a widely discussed subject, i.e., gut microbiome [172]. They explore gut microbiome influence on CAR-T cell therapy response and CRS. Their findings include *Bifidobacterium* correlation with stimulated production of inflammatory cytokines, such as IL-1 and IL-6, and subsequent possible association with CRS severity [173].

There is little knowledge on gut microbes’ ways of modulating the host’s immune system. Above all, bacterial communities are said to release intermediary metabolites and, in consequence, affect the host’s defenses [174]. Due to the contribution into innate and adaptive immunity of their host, gut microbiota are speculated to have an indirect impact on clinical outcomes of CAR-T cell therapy and grade of CRS [175].

#### 3.2.5. Race, Ethnicity and Obesity

In their article, Faruqi et al. observed no meaningful connections of race, ethnicity, or BMI to CAR-T consequences or OS [176]. It is only mentioned that Hispanic patients have higher risks for undergoing severe CRS. It is highlighted that CAR-T cell therapy is important for Hispanic and obese patients in particular, as they are more susceptible to the development of chemotherapy-resistant or refractory disease.

#### 3.2.6. Tumor Burden, Inflammation and Attributes of Axi-Cel

Tumor burden (TB) and inflammation have been studied as contributors for outcomes in patients with LBCL [177]. In the publication, the authors noted a few correlations, including high TB linked to lower probability of durable response and proinflammatory markers diminishing impact on the ratio of durable response and undermining in vivo expansion. Furthermore, product adequacy and amount of CD8 and CCR7+CD45RA+ T cells were observed to influence effectiveness of CAR-T cell therapy. Finally, TB, host inflammatory markers, the axi-cel specifics determined effectiveness and toxicity.

## 4. CAR-T Cell Therapy in Hodgkin lymphoma

Although the manuscript analyses CAR-T cell therapy in pediatric NHL, it should be mentioned that there are only a few studies about the use of CAR-T cell therapy in the treatment of Hodgkin lymphoma (HL) which is the most common cancer among adolescent young adult patients (aged 15–19). CAR-T cell therapy may be crucial, because there are still approximately 15% of all patients with R/R HL for whom first-line therapy (high-dose chemotherapy autologous stem cell transplantation) is ineffective. A Ramos’s study conducted among 41 patients aged 17–69 showed that CD30-targeted CAR-T cell therapy may be promising in the NH treatment. The ORR was 72%, with as many as 59% of patients achieving CR. There were also a high rate of durable responses and a good safety profile. Unfortunately, clinical trial results assessing the efficiency of CAR-T therapy in pediatric HL are not available [178,179].

## 5. Conclusions

Immunotherapy based on CAR-T cells seems to be a promising strategy for the treatment of non-Hodgkin lymphoma among the pediatric population. Due to the ongoing development of new CAR-T cell drugs that target a wide range of B-cell markers, more individualized treatment strategies are possible, especially for treatment-resistant patients. However, some challenges must also be overcome for the widespread use of CAR-T cells in children and adolescents. Because of the possible adverse events of CAR-T cell therapy, it is necessary to further develop the treatment strategies and inclusion criteria for patients. The significant problem is the long time and costs of producing drugs based on CAR-T cells. In addition, there is also a possibility that the patient may develop resistance to this type of immunotherapy. Additionally, available clinical trial results on the effectiveness and safety of CAR-T cell therapy in the treatment of R/R NHL mainly concern the adult population. However, many studies are currently being conducted on children and adolescents, which may prove to be a breakthrough in the treatment of NHL in this age group.

## Figures and Tables

**Figure 1 jpm-13-01595-f001:**
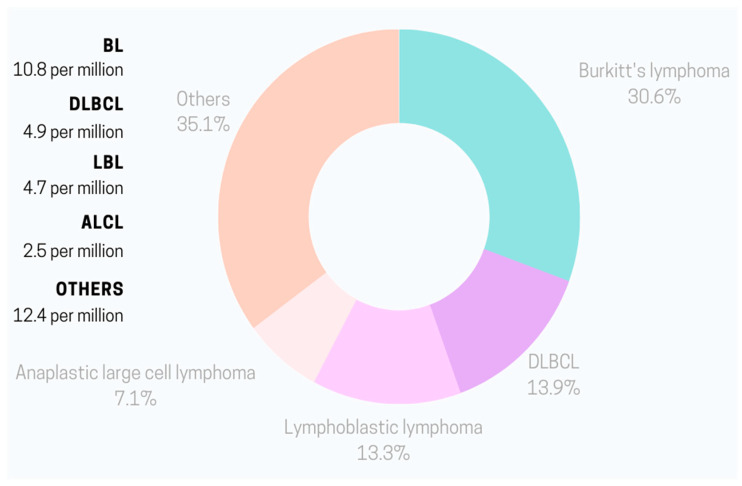
Pediatric non-Hodgkin lymphoma in children between 0 and 14 years old (both genders).

**Figure 2 jpm-13-01595-f002:**
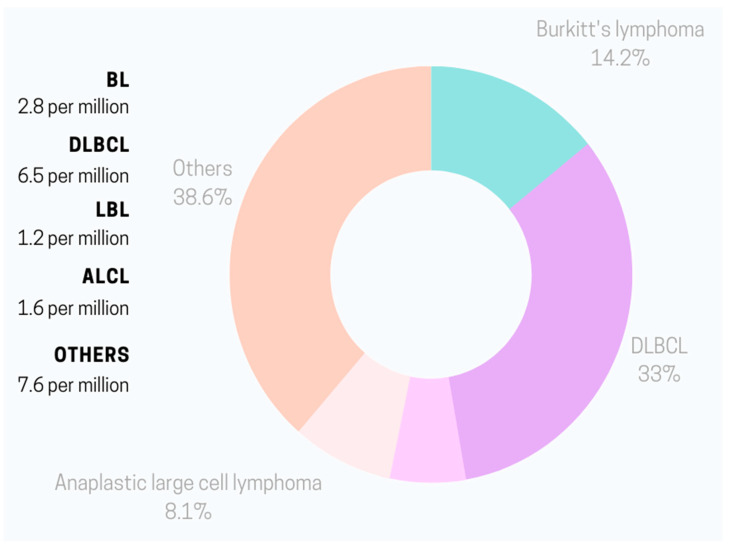
Pediatric non-Hodgkin lymphoma in adolescents between 15 and 19 years old (both genders).

**Table 1 jpm-13-01595-t001:** The main types of pediatric non-Hodgkin lymphoma.

WHO Classification	Immunophenotype	Chromosome Abnormalities	Affected Genes	Clinical Display
Burkitt’s lymphoma	Mature B-cell	t(8;14)(q24;q32), t(2;8)(p11;q24), t(8;22)(q24;q11)	*MYC*, *TCF3*, *ID3, CCND3*, *TP53*	Intra-abdominal (sporadic), head and neck (non-jaw, sporadic), jaw (endemic), bone marrow, CNS
DLBCL	Mature B-cell	No regular cytogenetic abnormality identified	*BCL2*, *MYC*, *SOCS1*, *KMT2D*	Nodal, abdominal, bone, primary CNS (when associated with immunodeficiency), mediastinal
T-lymphoblastic lymphoma	T lymphoblasts (TdT, CD2, CD3, CD7, CD4, CD8)	Loss of heterozygosity at 6q, deletions of CDKN2A in 9p21	*NOTCH1, FBXW7*, *CDKN2A*, *PTEN*, *KMT2D*	Mediastinal mass, bone marrow
B-lymphoblastic lymphoma	B lymphoblasts (CD19, CD79a, CD22, CD10, TdT)	Gene deletions	*CDKN2A, IKZF1*, and *PAX5*	Skin, soft tissue, bone, lymph nodes, bone marrow
Anaplastic large cell lymphoma	T-cell or null-cell expressing CD30	t(2;5)(p23;q35); less common variant translocations involving ALK	*ALK, NPM*	Generalized lymphadenopathy, bone marrow in 25%
Primary mediastinal (thymic) large B-cell lymphoma	Mature B cell, often CD30+	9p and 2p gains	*CIITA, TNFAIP3, SOCS1, PTPN11, STAT6*	Usually mediastinal; may also present with other nodal or extranodal disease

**Table 2 jpm-13-01595-t002:** The approved CAR-T drugs to treat B-cell non-Hodgkin lymphoma in adult patients.

Drug	Other Terms	Construction	Dose
Axicabtagene ciloleucel	axi-cel, Yescarta, KTE-CD19	CD19-CD28-CD3ζ	2 × 10^6^ cell per kg
Tisagenlecleucel	tisa-cel, Kymriah, CTL019	CD19–4-1BB-CD3ζ	(0.1–6) × 10^8^ cells
Brexucabtagene autoleucel	Tecartus, KTE-X19	CD19-CD28-CD3ζ	2 × 10^6^ cell per kg
Lisocabtagene maraleucel	liso-cel, Breyanzi, JCAR017	CD19–4-1BB-CD3ζ	50 × 10^6^, 100 × 10^6^ and 150 × 10^6^ cells

**Table 3 jpm-13-01595-t003:** Recruiting clinical trials assessing the use of CAR-T cell therapies in children and adults with NHLs.

Drug	ClinicalTrials.gov Identifier	Phase of Clinical Study	Estimated Numbers of Patients	Studied Patient Population (Diagnose, Age)	Dosage	References
CD19 CAR-T cells	NCT04532203	Early phase 1	72	CNS Involvement of R/R CD19+ B-NHL/B-ALL, 3–75 y.o.	-	[66]
CD19 CAR-T cells (SAGAN)	NCT01853631	Phase 1	64	CD19+ BCL/leukemia ≤ 75 y.o.	Three dose levels— DL1: 1 × 10^6^ cells/m^2^, DL2: 5 × 10^6^ cells/m^2^, DL3: 2 × 10^7^ cells/m^2^	[67]
murine CD19 CAR-T cells	NCT04532281	Early phase 1	120	R/R CD19+ B-NHL/B-ALL, no age restrictions	-	[68]
humanized CD19 CAR-T cells	NCT04532268	Early phase 1	72	R/R CD19+ B-NHL/B-ALL, 3–75 y.o.	-	[69]
CD19 CAR-T cells	NCT04271410	Phase 1/2	80	R/R CD19+ B-NHL/B-ALL/B-CLL, 2–75 y.o.	-	[70]
CD19 CAR-T cells	NCT02935257	Phase 1	60	R/R CD19+ DLBCL/FL/MCL/B-ALL/CLL/SLL, ≥16 y.o.	-	[71]
CD19 CAR-T cells	NCT06027957	Phase 1	16	R/R CD19+ NHL/ALL, 1–60 y.o.	1–2 × 10^6^ cells/kg	[72]
CD19 CAR-T cells	NCT03938987	Phase 1b/2	63	R/R CD19+ NHL/ALL, 2–70 y.o.	Three dose levels— DL1: 0.5 × 10^6^ cells/kg, DL2: 1.0 × 10^6^ cells/kg, DL3: 2.0 × 10^6^ cells/kg	[73]
CD19 CAR-T cells (PL001)	NCT05326243	Phase 1/2	49	CD19+ DLBCL/PMLBCL/FL, 14–70 y.o.	0.1–9 × 10^6^ cells/kg	[74]
CD19 CAR-T cells	NCT05143112	Phase 1/2	20	R/R CD19+ B-NHL, 14–70 y.o.	-	[75]
CD19 CAR-T cells (WZTL-002)	NCT04049513	Phase 1	30	R/R CD19+ B-NHL, 16–75 y.o.	Four dose levels— DL1: 5 × 10^4^ cells/kg, DL2, DL3, DL4: not specified	[76]
CD19 CAR-T cells	NCT04787263	Phase 1/2	32	R/R CD19+ DLBCL/PMLBCL/B-ALL, 1–25 y.o.	1–3 × 10^6^ cells/kg	[77]
CD19 CAR-T cells	NCT04544592	Phase 1/2	50	R/R CD19+ B-NHL/B-ALL, 30 days–25 y.o.	-	[78]
CD19 CAR-T cells (CT-RD06)	NCT04226989	Early phase 1	72	R/R CD19+ B-NHL/B-ALL, 3–70 y.o.	-	[79]
CD19 CAR-T cells	NCT03853616	Phase 1/2	48	R/R CD19+ B-NHL/B-ALL/B-CLL, ≥1 y.o.	Four dose levels— DL0: 1 × 10^5^ cells/kg, DL1: 5 × 10^5^ cells/kg, DL2: 1 × 10^6^ cells/kg, DL3: 3 × 10^6^ cells/kg	[80]
CD19 CAR-T cells (Brexucabtagene Autoleucel, KTE-X19) (ZUMA4)	NCT02625480	Phase 1/2	116	R/R B-NHL/B-ALL, ≤21 y.o.	Two dose levels— DL1: 1 × 10^6^ cells/kg, DL2: 2 × 10^6^ cells/kg	[36]
CD19 CAR-T cells (JCAR017)	NCT03743246	Phase 1/2	121	R/R CD19+ B-NHL/B-ALL, ≤25 y.o.	Five dose levels— DL1: 0.05–0.75 × 10^6^ cells/kg, DL2, DL3, DL4, DL5: not specified	[81]
CD19 CAR-T cells/chidamide bridging + CD19 CAR-T cells	NCT05370547	Phase 1/2	120	R/R CD19+ B-NHL, 16–75 y.o.	-	[82]
allo-HSCT + CD19 CAR-T cells	NCT02050347	Phase 1	40	CD19+ B-NHL/B-ALL/B-CLL, no age restrictions	Three dose levels— DL1: 1 × 10^5^ cells/kg, DL2: 5 × 10^5^ cells/kg, DL3: 1 × 10^6^ cells/kg (dose escalation 1) or DL1: 5 × 10^5^ cells/kg, DL2: 1 × 10^6^ cells/kg, DL3: 5 × 10^6^ cells/kg (dose escalation 2)	[83]
CD19 CAR-NKT cells (ANCHOR)	NCT03774654	Phase 2	48	R/R CD19+ B-NHL/B-ALL/B-CLL 3–75 y.o.	Three dose levels— DL1: 1 × 10^7^/m^2^, DL2: 3 × 10^7^/m^2^, DL3: 1 × 10^8^/m^2^	[84]
CD19 CAR-NKT cells (KUR-502) (ANCHOR2)	NCT05487651	Phase 1	36	R/R CD19+ B-NHL/B-ALL/B-CLL, 3–75 y.o.	Three dose levels— DL1: 1 × 10^7^ cells/m^2^, DL2: 3 × 10^7^ cells/m^2^, DL3: 1 × 10^8^ cells/m^2^	[85]
CD19/CD22 CAR-T cells	NCT05098613	Phase 1/1b	20	R/R B-NHL/HL, ≥16 y.o.	-	[86]
CD19/CD22 CAR-T cells (SL19+22)	NCT05206071	Not applicable	100	R/R CD19+ and/or CD22+ NHL, 3–75 y.o.	-	[87]
CD19/CD22 CAR-T cells	NCT04715217	Phase 1/2	24	R/R CD19+ and/or CD22+ BCL, 6–70 y.o.	Three dose levels— DL1: 0.5 × 10^6^ cells/kg, DL2: 1 × 10^6^ cells/kg, DL3: 2 × 10^6^ cells/kg	[88]
CD19/CD22 CAR-T cells	NCT04782193	Phase 1/2	40	R/R CD19+ and/or CD22+ BCL, 2–75 y.o.	-	[89]
CD19/CD22 CAR-T cells	NCT04648475	Phase 1/2	40	R/R CD19+ and CD22+ BCL/leukemia, 3–75 y.o.	-	[90]
CD19/CD22 CAR-T cells	NCT04649983	Phase 1/2	40	R/R CD19+ and CD22+ BCL/leukemia, 2–75 y.o.	-	[91]
CD19/CD22 CAR-T cells	NCT03448393	Phase 1	140	R/R CD19+/CD22+ B-NHL/B-ALL/isolated CNS ALL, 3–39 y.o.	Five dose levels— DL1: 1 × 10^5^ cells/kg, DL2: 3 × 10^5^ cells/kg, DL3: 1 × 10^6^ cells/kg, DL4: 3 × 10^6^ cells/kg, DL5: 1 × 10^7^ cells/kg	[92]
CD19/CD22/CD30/CD7/CD79 CAR-T cells	NCT04666168	Not applicable	200	R/R CD19+ or CD22+/CD30+/CD7+/ CD79+ NHL, 14–75 y.o.	-	[93]
CD19/CD20/CD22/CD70/PSMA/ CD13/CD79b/GD2 CAR-T cells	NCT04429438	Phase 1/2	11	CD19+ and/or CD22+/CD70+/PSMA+/CD13+/CD79b+/GD2+ PMBCL/ BCL involving CNS, 6 mths–75 y.o.	-	[94]
CD30 CAR-T cells	NCT02917083	Phase 1	60	R/R CD30+ NHL/HL, 12–75 y.o.	-	[95]
CD30 CAR-T cells	NCT03383965	Phase 1	20	CD30+ ALCL/HL, 2–80 y.o.	-	[96]
CD30 CAR-T cells	NCT02259556	Phase 1/2	40	CD30+ NHL/HL, 16–18 y.o.	-	[97]
CD30 CAR-T cells	NCT02690545	Phase 1/2	40	R/R CD30+ NHL/HL, 3–17 y.o.	Two dose levels— DL1: 1 × 10^8^ cells/m^2^, DL2: 2 × 10^8^ cells/m^2^	[98]
CD30 CAR-EBVST cells (Epstein–Barr virus-specific T cells)	NCT04288726	Phase 1	18	CD30+ DLBCL/NK/TL/HL, 12–75 y.o.	Three dose levels— DL1: 4 × 10^7^ cells/m^2^, DL2: 1 10^8^ cells/m^2^, DL3: 4 × 10^8^ cells/m^2^	[99]
CD5 CAR-T cells	NCT04594135	Phase 1	20	R/R T-LLy/T-ALL, ≥ 8 y.o.	-	[100]
CD5 CAR-T cells (MAGENTA)	NCT03081910	Phase 1	42	R/R CD5+ T-ALL/T-LLy/T-NHL, ≤ 75 y.o.	Three dose levels— DL1: 1 × 10^7^ cells/m^2^, DL2: 5 × 10^7^ cells/m^2^, DL3: 1 × 10^8^ cells/m^2^	[101]
CD7 CAR-T cells	NCT05290155	Phase 1	4	R/R CD7+ T-LLy/T-ALL, 14–70 y.o.	Three dose levels— DL1: 0.5 × 10^6^ cells/kg, DL2: 2 × 10^6^ cells/kg, DL3: 5 × 10^6^ cells/kg	[102]
CD7 CAR-T cells	NCT03690011	Phase 1	21	R/R CD7+ T-NHL/CTCL/T-ALL, ≤75 y.o.	Three dose levels— DL1: 1 × 10^7^ cells/m^2^, DL2: 3 × 10^7^ cells/m^2^, DL3: 1 × 10^8^ cells/m^2^	[103]
CD7 CAR-T cells (ThisCART7)	NCT05127135	Phase 1	30	R/R CD7+ T-NHL/T-ALL/T-LBL, 3–70 y.o.	0.5–6 × 10^6^ cells/kg	[104]
CD7 CAR-T cells	NCT04823091	Phase 1	24	R/R CD7+ T-NHL/T-CLL, 14–70 y.o.	Two dose levels— DL1: 1 × 10^6^ cells/kg, DL2: 2 × 10^6^ cells/kg	[105]
CD70 CAR T-cells	NCT04662294	Early phase 1	108	R/R CD70+ T-NHL/T-ALL/AML/MM, no age restrictions	-	[106]
CAR-T cells	NCT05619861	Not applicable	20	R/R Hematopoietic and lymphatic tumor, 14–75 y.o.	0.1–3 × 10^6^ cells/kg	[107]
Igβ CAR-T cells	NCT05312476	Phase 2	12	R/R Igβ+ NHL, ≥6 y.o.	Three dose levels— DL1: 1 × 10^6^ cells/kg, DL2: 3 × 10^6^ cells/kg, DL3: 6 × 10^6^ cells/kg	[108]

Abbreviations: ALCL—anaplastic large cell lymphoma; AML—acute myeloid leukemia; B-ALL—B-cell acute lymphoblastic leukemia; BCL—B-cell lymphoma; B-CLL—B-cell chronic lymphocytic leukemia; B-NHL—B-cell non-Hodgkin lymphoma; CNS—central nervous system; CTCL—cutaneous T-cell lymphoma; DLBCL—diffuse large B-cell lymphoma; DL—dose level; FL—follicular lymphoma; HL—Hodgkin lymphoma; MM—multiple myeloma; MCL—mantle cell lymphoma; NK/TL—natural killer/T-cell lymphoma; PMLBCL—primary mediastinal large B-cell lymphoma; R/R—relapsed/refractory; SLL—small lymphocytic lymphoma; T-ALL—T-cell acute lymphoblastic leukemia; T-CLL—T-cell chronic lymphocytic leukemia; T-LBL/T-LLy—T-cell lymphoblastic lymphoma; T-NHL—T-cell non-Hodgkin lymphoma; y.o.—years old.

## Data Availability

No new data were created or analyzed in this study. Data sharing is not applicable to this article.

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
