# Peer review of "CAR-T Cell Therapy in the Treatment of Pediatric Non-Hodgkin Lymphoma"

_jpm, 2023, doi:10.3390/jpm13111595_

Round 1

Reviewer 1 Report

Comments and Suggestions for Authors

Thank you for the authors for this interesting paper.

However, some clarifications are needed.

1. Table 1 has * in the sections of Burkitt-like lymphoma and Large B-cell lymphoma with IRF4 reaarregement, but no explanation for the asterix. Please add this. The table is Table in English (now Tabela 1-3). Clinical Display is more commonly presented as Common affisions (or something like that). Please add some information also to empty sections. What is the difference in incidence between common and rare lymphomas?

2. As pediatric cancer is rare for me as an adult oncologist the line 47 sentence: ...approximately 45 % of all lymphomas... is not plausible. There are multiple numbers of adults (age range 18-120, not only 0-17) and the incidence is much more in adults. Please recheck this issue.

3. Figure 3. 15-19 years old are not children, but rather adolescents. Please change the term.

4. Line 131, 313, etc.: Side effects are usually adverse events. Please change the term.

5.  The title of the paper is Pediatric use of CAR-T therapy. Thus the text from line 119 forward should be consolidated a lot and rethought to present only the important issues for this title. A table perhaps? Please notice the next questions in this review:

6. Line 272: Does that mean: to achieve a CR or PR was 1 month? Please clarify.

7. The clinical significance of staying in CR or PR differs markedly. These should be separated when writing the new text (for example line 311).

8. The Discussion should be expanded and should include a paragraph about if there are any evidence or suspicions that these treatment results or adverse events differ from adults, future time line for these treatments, etc.

Comments on the Quality of English Language

Quality of language was poor and should be corrected.

Reviewer 2 Report

Comments and Suggestions for Authors

This is a very large review on different CAR T cell therapy used in Non-Hodgkin lymphoma.   CAR T efficacy and toxicity have been reported whether or not they were on the market. The authors tried to make relation with the B cell  target  of the different subtypes.

Table 1:quite incomplete !  immunophenotype can be added ex: anaplastic lymphoma is CD30 + (and responding to brentuximab vedotin)

I suggest to restrict to the main subtypes .

Most of the studies have been performed and registered until now in adult lymphoma patients.

In the different paragraphs, it is not clear if the results observed with CAR-T described for adults are available for children. Clarification could be made for  each type of CAR T with a paragraph providing number and type of lymphoma with response and survival .

Table3: the title is” in children”, obviously you include adults patients .  You can take this opportunity to provide the number of the subset of children in the different studies.  

Round 2

Reviewer 1 Report

Comments and Suggestions for Authors

The corrections were made as asked, although the paper is still quite long. This is left to be decided to the editors of the journal.

Reviewer 2 Report

Comments and Suggestions for Authors

The authors answered to the comments with the limits of the data available

The manuscript has been improved
